# Validation of Prognostic Stage and Anatomic Stage in the American Joint Committee on Cancer 8th Edition for Inflammatory Breast Cancer

**DOI:** 10.3390/cancers12113105

**Published:** 2020-10-24

**Authors:** Kumiko Kida, Kenneth R. Hess, Bora Lim, Toshiaki Iwase, Sudpreeda Chainitikun, Vicente Valero, Anthony Lucci, Huong Carisa Le-Petross, Wendy A. Woodward, Savitri Krishnamurthy, Gabriel N. Hortobagyi, Debu Tripathy, Naoto T. Ueno

**Affiliations:** 1Morgan Welch Inflammatory Breast Cancer Research Program and Clinic, The University of Texas MD Anderson Cancer Center, Houston, TX 77030, USA; kidakumiko117@gmail.com (K.K.); BLim@mdanderson.org (B.L.); TIwase@mdanderson.org (T.I.); SChainitikun@mdanderson.org (S.C.); (vvalero@mdanderson.org (V.V.); alucci@mdanderson.org (A.L.); hlepetross@mdanderson.org (H.C.L.-P.); wwoodward@mdanderson.org (W.A.W.); skrishna@mdanderson.org (S.K.); 2Department of Breast Medical Oncology, The University of Texas MD Anderson Cancer Center, Houston, TX 77030, USA; ghortoba@mdanderson.org (G.N.H.); DTripathy@mdanderson.org (D.T.); 3Department of Biostatistics, The University of Texas MD Anderson Cancer Center, Houston, TX 77030, USA; khess@mdanderson.org; 4Department of Breast Surgical Oncology, The University of Texas MD Anderson Cancer Center, Houston, TX 77030, USA; 5Department of Diagnostic Radiology, The University of Texas MD Anderson Cancer Center, Houston, TX 77030, USA; 6Department of Radiation Oncology, The University of Texas MD Anderson Cancer Center, Houston, TX 77030, USA; 7Department of Pathology, The University of Texas MD Anderson Cancer Center, Houston, TX 77030, USA

**Keywords:** inflammatory breast cancer, the American Joint Committee on Cancer, staging, anatomic stage, prognostic stage

## Abstract

**Simple Summary:**

This study is the first to confirm that the clinical prognostic stage in the American Joint Committee on Cancer (AJCC) 8th edition provides a more accurate prognostication for inflammatory breast cancer than the traditional anatomic stage. It indicated that incorporating biological factors into the traditional staging system provides more accurate inflammatory breast cancer (IBC) prognosis stratification than does the tumor size (T) and presence of lymph node involvement (N) and distant metastasis (M) (TNM) staging system. Our results show that the AJCC prognostic staging system is optimal for prognostication in IBC.

**Abstract:**

The AJCC updated its breast cancer staging system to incorporate biological factors in the “prognostic stage”. We undertook this study to validate the prognostic and anatomic stages for inflammatory breast cancer (IBC). We established two cohorts of IBC diagnosed without distant metastasis: (1) patients treated at The University of Texas MD Anderson Cancer Center between 1991 and 2017 (MDA cohort) and (2) patients registered in the national Surveillance, Epidemiology, and End Results (SEER) database between 2010 and 2015 (SEER cohort). For prognostic staging, estrogen receptor (ER)+/progesterone receptor (PR)+/ human epidermal growth factor receptor-2 (HER2)+/grade 1–2 was staged as IIIA; ER+/PR−/HER2−/grade 3, ER−/PR+/HER2−/grade 3, and triple-negative cancers as IIIC; and all others as IIIB. Endpoints were breast cancer-specific survival (BCSS), overall survival (OS), and disease-free survival (DFS). We studied 885 patients in the MDA cohort and 338 in the SEER cohort. In the MDA cohort, the prognostic stage showed significant predictive power for BCSS, OS, and DFS (all *p* < 0.0001), although the anatomic stage did not. In both cohorts, the Harrell concordance index (C index) was significantly higher in the prognostic stage than the anatomic stage for all endpoints. In conclusion, the prognostic stage provided more accurate prognostication for IBC than the anatomic stage. Our results show that the prognostic staging is applicable in IBC.

## 1. Introduction

The central role of a cancer staging system is to predict patients’ survival. The standard cancer staging system is the American Joint Committee on Cancer (AJCC) staging system, which assigns stage based on tumor size (T) and the presence of lymph node involvement (N) and distant metastasis (M). However, this traditional TNM staging system is insufficient for predicting breast cancer patients’ survival because it omits important biological factors, such as hormone receptor (HR) and human epidermal growth factor receptor-2 (HER2) status, which have significant impact on survival. Thus, the AJCC recently updated its breast cancer staging system by including a “prognostic stage” that accounts for biological factors.

The updated AJCC Cancer Staging Manual, 8th edition [1], includes anatomic and prognostic staging. The anatomic stage is the traditional staging system defined by TNM categories. The prognostic stage is based on TNM categories and four biological factors: tumor grade, estrogen receptor (ER), progesterone receptor (PR), and HER2 status. The prognostic stage, developed using data from patients registered in the National Cancer Database and treated between 2010 and 2011 [1,2], has been validated and proven to have survival prediction power in breast cancer [3,4,5]. 

Inflammatory breast cancer (IBC), an extremely aggressive breast cancer subgroup, is clinically defined as diffuse erythema and edema of the breast in the presence of pathologic evidence of breast cancer [1,6]. IBC patients’ prognosis is poor, and prognostic markers are controversial [7,8]. The traditional anatomic staging system assigns non-metastatic IBC as stage IIIB (for N0-2 nodal status) or IIIC (for N3), whereas the prognostic staging system assigns stages IIIA-C according to biological factors, not nodal status [1]. Neither staging system has been validated in IBC, which was excluded from previous validation studies because of its unique pathologic nature. Therefore, it is critical to validate the staging systems for IBC. 

The purpose of this study was to validate the prognostic predictive power of the AJCC’s prognostic and anatomic stages for IBC using two independent, large-population databases. To quantify the staging system’s predictive power, we applied the Harrell concordance index (C index). A higher C index value indicated better predictive power of the staging system.

## 2. Results

### 2.1. Patient and Staging Characteristics

We identified 1045 patients with non-metastatic IBC treated between January 1991 and January 2017 in our institutional database; 160 (15.3%) were excluded because of incomplete staging records, leaving 885 patients eligible for the MDA cohort. Patient characteristics are shown in Table 1. The anatomic stage was IIIB in 662 patients (74.8%) and IIIC in 223 patients (25.2%). The prognostic stage was IIIA in 21 patients (2.4%), IIIB in 524 patients (59.2%), and IIIC in 340 patients (38.4%). Compared with the anatomic stage, the prognostic stage upstaged 248 patients (28.0%) and downstaged 146 (16.5%). Within the MDA cohort, 140 patients (15.8%) went down one stage (from IIIB to IIIA or IIIC to IIIB) and 6 (0.7%) went down two stages (from IIIC to IIIA).

In the SEER database, we identified 601 patients with non-metastatic IBC registered between 2010 and 2015. We excluded 263 patients (43.8%) with incomplete staging records, leaving 338 patients eligible for the SEER cohort. The anatomic stage was IIIB in 271 patients (80.2%) and IIIC in 67 patients (19.8%). The prognostic stage was IIIA in 16 patients (4.7%), IIIB in 216 patients (63.9%), and IIIC in 106 patients (31.4%). Compared with the anatomic stage, the prognostic stage upstaged 84 patients (24.9%) and downstaged 58 (17.2%). Within the SEER cohort, 55 patients (16.3%) went down one stage and 3 (0.9%) went down two stages.

### 2.2. Survival Analysis

The median length of follow-up was 6.7 years (95%CI, 6.3–7.2) for the MDA cohort and 3.3 years (95%CI, 3.0–3.7) for the SEER cohort using reverse Kaplan-Meier methods. In the MDA cohort, 464 deaths (52.4%) were observed, including 425 patients (48.0%) who died of breast cancer. In the SEER cohort, 90 deaths (26.6%) were observed, including 61 patients (18.0%) who died of breast cancer. 

Figure 1 shows survival outcomes in the MDA cohort by anatomic and prognostic stages. By the anatomic stage, no statistically significant differences were observed for BCSS, OS, and DFS (Figure 1A). In contrast, by prognostic stage, there were significant differences for all endpoints (Figure 1B). The hazard ratio for prognostic stage IIIC vs. IIIB was 2.35 for BCSS (95% CI, 1.94–2.84; *p* < 0.0001), 2.32 for OS (95% CI, 1.93–2.79; *p* < 0.0001), and 1.84 for DFS (95% CI, 1.55–2.20; *p* < 0.0001). The C index was significantly higher in the prognostic stage than the anatomic stage for BCSS (0.643 vs. 0.526, *p* < 0.0001), OS (0.641 vs. 0.533, *p* < 0.0001), and DFS (0.599 vs. 0.494, *p* < 0.0001). Stage IIIA patients had the best survival rates with the 5-year BCSS of 83%. However, the significance of the difference between prognostic stages IIIA and IIB could not be determined because, with only 21 stage IIIA patients, statistical power was insufficient.

Figure 2 shows survival outcomes in the SEER cohort by anatomic and prognostic stages. Statistically significant difference was observed in BCSS and OS by anatomic stage (Figure 2A). The hazard ratio for anatomic stage IIIC vs. IIIB was 1.80 for BCSS (95% CI, 1.04–3.12; *p* = 0.04) and 1.97 for OS (95% CI, 1.26–3.07; *p* = 0.003). The survival differences between IIIB and IIIC appeared more prominent with the prognostic stage than with the anatomic stage (Figure 2B). The hazard ratio for prognostic stage IIIC vs. IIIB was 3.60 for BCSS (95% CI, 2.16–6.00; *p* < 0.0001) and 3.26 for OS (95% CI, 2.14–4.94; *p* < 0.0001). The C index was significantly higher in the prognostic stage than it was in the anatomic stage for BCSS (0.685 vs. 0.550, *p* = 0.0028) and OS (0.681 vs. 0.557, *p* = 0.0008). As in the MDA cohort, stage IIIA patients had the best survival rates, and no events were observed in this group. 

In both cohorts, prognostic stage IIIC had the worst prognosis. It consisted of three subpopulations: (1) ER+/PR−/HER2−/grade 3 (*n* = 71 in the MDA cohort, 19 in the SEER cohort); (2) ER−/PR+/HER2−/grade 3 (*n* = 20 in the MDA cohort, 4 in the SEER cohort); and (3) triple-negative (*n* = 249 in the MDA cohort, 83 in the SEER cohort). Of these subpopulations, triple-negative IBC is known to have the worst prognosis, but the prognosis for the other two subgroups is not well known. To determine whether all three subpopulations were valid for prognostic stage IIIC, we compared the survival data of the prognostic stage IIIC subpopulations to those for prognostic stage IIIB. Figure 3 and Figure 4 show the BCSS and OS results, respectively. The prognoses for ER+/PR−/HER2−/grade 3 and ER−/PR+/HER2−/grade 3 were better than that for triple-negative IBC and worse than that for prognostic stage IIIB. The hazard ratio for ER+/PR−/HER2−/grade 3 vs. prognostic stage IIIB was 1.84 for BCSS (95% CI, 1.30–2.61; *p* = 0.0010) and 1.93 for OS (95% CI, 1.40–2.67; *p* < 0.0001) in the MDA cohort, and 2.36 for BCSS (95% CI, 0.90–6.18; *p* = 0.080) and 2.72 for OS (95% CI, 1.32–5.61, *p* = 0.007) in the SEER cohort. The hazard ratio for ER−/PR+/HER2−/grade 3 vs. prognostic stage IIIB was 0.94 for BCSS (95% CI, 0.48–1.85; *p* = 0.87) and 1.03 for OS (95% CI, 0.56–1.89, *p* = 0.93) in the MDA cohort, and 2.32 for BCSS (95% CI, 0.31–17.14; *p* = 0.41) and 1.52 for OS (95% CI, 0.21–11.1; *p* = 0.68) in the SEER cohort.

## 3. Discussion

This study investigated the survival predictive power of the IBC clinical prognostic and anatomic stages in the AJCC’s 8th edition staging system. Our study is the first to confirm that the prognostic stage provides a more accurate prognostication for IBC than the anatomic stage in both a single-institution cohort and a national registry. The results indicated that incorporating biological factors into the traditional staging system provides more accurate IBC prognosis stratification than does the TNM staging system.

For IBC, the AJCC’s anatomic staging stratifies only patients with N3 status as stage IIIC. All other patients with non-metastatic IBC are staged as IIIB. The AJCC defines clinical N3 as metastasis in (1) one or more ipsilateral infraclavicular lymph nodes or ipsilateral supraclavicular lymph nodes, or (2) clinically detected ipsilateral internal mammary lymph nodes and clinically evident axillary lymph nodes [1]. Although N3 status is generally a poor prognostic factor for non-IBC breast cancer, no previous studies have validated the impact of N3 status on IBC prognosis. In this study, we demonstrated that anatomic stage was statistically significant for IBC prognosis in the SEER cohort, but not the MDA cohort. This result implies that metastasis in infra-/supraclavicular or internal mammary lymph nodes is unclear as an IBC prognostic factor. 

The new prognostic stage provides three staging groups, IIIA–IIIC, on the basis of biological factors. Prognostic stage IIIA includes ER+/PR+/HER2+/grade 1–2 cancers. One notable observation in the present study was the good prognosis for stage IIIA, even though there were no significant differences in survival outcomes between stage IIIA and other stages due to stage IIIA’s small sample size. We found no breast cancer-specific deaths for stage IIIA in the SEER cohort. Most of the four previous studies comparing survival in HR+ (ER+ and/or PR+)/HER2+ vs. other subtypes have demonstrated that the HR+/HER2+ subtype has the best survival outcomes, but these studies did not have sufficient statistical power to demonstrate significant differences [9,10,11,12]. These results were compatible with ours.

In our study, prognostic stage IIIC, which included (1) ER+/PR−/HER2−/grade 3, (2) ER−/PR+/HER2−/grade 3, and (3) triple-negative IBC, had significantly worse prognoses than other stages in both cohorts. Six previous studies investigating the relationship between subtypes and survival outcomes in IBC have reported that triple-negative IBC has the worst survival outcomes [9,10,11,12,13,14]. These studies supported our findings that IBC’s prognostic stage IIIC has a poor prognosis because the triple-negative type is the major component of prognostic stage IIIC. However, an important question is whether the ER+/PR−/HER2−/grade 3 and ER−/PR+/HER2−/grade 3 subpopulations actually have poor prognoses and should be included in prognostic stage IIIC. In our current study, the survival curves for these two subpopulations were between those for the triple-negative type and stage IIIB (Figure 3 and Figure 4). Therefore, we consider it appropriate to include these subpopulations in prognostic stage IIIC. 

It is important to note that, between the cohorts, there were three major differences in characteristics that might have influenced the survival outcomes. First, the median follow-up duration was 6.7 years for the MDA cohort and 3.3 years for the SEER cohort. Second, we enrolled patients treated between 1991 and 2017 for the MDA cohort and between 2010 and 2015 for the SEER cohort because the SEER database did not include HER2 status before 2010. This difference in timeframe might have created discrepancies in patients’ treatment regimens, especially regarding anti-HER2 therapy. However, we could not compare treatment details because the SEER database lacks treatment information. Third, 82% of MDA cohort patients received trimodality therapy (chemotherapy, surgery, and radiation therapy), but we do not have such information for the SEER cohort. We have previously reported that only 60–70% of IBC patients in the National Cancer Database received trimodality therapy and that its underutilization negatively affected IBC patients’ survival [15]. Therefore, whether or not patients received trimodality therapy could have been a confounding factor for survival outcomes. Overall, these three differences in patient characteristics may account for the discrepancy between the two cohorts in whether the anatomic stage stratified survival outcomes with statistical significance. Notably, despite these differences, the prognostic stage showed its prognostic power in both cohorts. 

This is the first study to validate the current AJCC staging system for patients with IBC. Even the traditional anatomic stage has never been validated for its survival predictive power in IBC. This study is the first to question the usefulness of the anatomic stage and to show the significant survival predictive power of the prognostic stage in IBC. Another strength of our study is that we had two independent, large cohorts, which is unusual because IBC is rare and large IBC databases are very limited. Our large cohorts enabled us to lessen bias and validate the staging system with greater accuracy. 

Several limitations of this study should be acknowledged. First, because of the retrospective nature of the study, the treatment histories were varied among patients. Second, information on systemic therapy, including the administration of trastuzumab, was unavailable for the SEER cohort. Third, although the overall sample size was large, the number of patients in some staging groups, such as stage IIIA, was too small for analysis, resulting in limited statistical power. Fourth, follow-up duration in the SEER cohort was limited relative to the long natural history of breast cancer. Fifth, while diagnostic criteria of IBC were unified in the MDA cohort, the criteria might not be consistent in the SEER cohort. Some of these limitations could not be avoided because IBC is a rare disease and existing large-population databases are very limited. Despite these limitations, our study demonstrated the clinical significance of the current staging system in IBC. 

Although our results showed that the prognostic stage is useful for IBC, the IBC staging system still has room for improvement. Because the prognostic stage was created using data from non-IBC patients, the staging design is not as well modeled for IBC. For example, the prognostic stage for IBC is defined only by biological factors and excludes lymph node status. Since aggressiveness and prognosis are distinct in IBC compared with non-IBC, a better staging system based on IBC data should be investigated. 

## 4. Materials and Methods 

### 4.1. Study Design and Data Source 

This retrospective study sought to validate the AJCC’s staging systems for inflammatory breast cancer (IBC) patients using two IBC patient cohorts: one from The University of Texas MD Anderson Cancer Center (MDA cohort) and the other from the national Surveillance, Epidemiology, and End Results (SEER) database (SEER cohort). 

For the MDA cohort, we identified consecutive IBC patients who had had their initial definitive surgery at our institution from January 1991 to January 2017. We selected patients evaluated at diagnosis, with confirmed IBC, and with documented follow-up data for survival. At MD Anderson Cancer Center, a multidisciplinary panel diagnosed IBC based on the diagnostic criteria described in the expert panel consensus statement [6].For the SEER cohort, we identified patients with IBC who were registered in the SEER database from 2010 (when SEER began recording HER2 status) to 2015 (the latest year in the database at the time of this study). We extracted all IBC cases in the SEER database using the histology code for inflammatory carcinoma (8530) from the International Classification of Diseases for Oncology, 3rd edition (ICD-O-3) [16]. For both cohorts, we selected cases with documented tumor grade, ER, PR, HER2, and vital status.

We excluded from both datasets: (1) Patients with stage IV disease; (2) patients who did not receive definitive surgery for IBC; (3) patients with secondary IBC (any IBC occurring after non-IBC presentation or noted as “secondary IBC” in the medical record) and/or suspected non-IBC; and/or (4) patients with incomplete records regarding clinical and pathological characteristics.

The institutional review board approved this study (protocol number PA18-0455). Informed consent was waived for this retrospective analysis.

### 4.2. Data Collection

Data evaluated included age at diagnosis; TNM categories; ER, PR, and HER2 status; histological subtype; tumor grade; cause-specific death classification; survival months; and vital status. Chemotherapy and radiotherapy data in the SEER database were not detailed enough to be assessed. For the MDA cohort, ER status was determined by immunohistochemistry and defined as positive with a cutoff of 10% before 2010 and 1% after 2010, when the American Society of Clinical Oncology/College of American Pathologists (ASCO/CAP) guideline changed [17]. HER2 status was defined as positive if scored as 3+ on immunohistochemistry or if fluorescence in situ hybridization demonstrated gene amplification, according to the ASCO/CAP guidelines [18]. In the SEER database, ER status has been traditionally categorized as positive (≥10% stained cells), negative (no stained cells), or borderline (1–9% stained cells) [19]. In this study, we counted borderline ER status as positive because the recent ASCO/CAP guideline listed 1% as the positive cutoff value. For tumor grades, grades 1, 2, and 3 corresponded to well, moderately, and poorly/undifferentiated tumors, respectively. 

The anatomic and prognostic stages were determined according to the AJCC Cancer Staging Manual, 8th edition [1]. We used the clinical stage determined prior to preoperative treatment because most patients received neoadjuvant chemotherapy. For the anatomic stage, all IBC patients were classified as T4d; those with N0-2 disease were staged as IIIB, and those with N3 disease were staged as IIIC. For prognostic stage, ER+/PR+/HER2+/grade 1–2 was staged as IIIA; ER+/PR−/HER2−/grade 3, ER−/PR+/HER2−/grade 3, and triple-negative cancers were staged as IIIC; and all others were staged as IIIB.

### 4.3. Survival Analysis 

The survival endpoints in this study were breast cancer-specific survival (BCSS), overall survival (OS), and disease-free survival (DFS). BCSS and OS were assessed in both cohorts. DFS was assessed only in the MDA cohort because cancer recurrence data were unavailable in the SEER database. All survival data were calculated from the date of diagnosis. The follow-up cutoff dates were 8 August 2018, for the MDA cohort, and 31 December 2015, for the SEER cohort. Any patient alive at those dates was censored at those times in survival analysis. For the BCSS analysis, patients who died from a cause other than breast cancer were censored on the date of death. The differences in each endpoint between staging groups were statistically compared according to the prognostic and anatomic staging systems. 

### 4.4. Statistical Considerations

The log-rank test was used to compare differences in endpoints between staging groups. The Harrell concordance index (C index) was used to quantify the staging models’ predictive performance. The C index measured the goodness of fit for binary outcomes in a logistic regression model. It gave the probability that a randomly selected patient who experienced an event had a higher risk score than a patient who had not experienced the event. A value below 0.5 indicated a very poor model and a value of 0.5 meant that the model was no better at predicting an outcome than random chance. A higher C index value indicated a better predictive performance. The hazard ratio was used to calculate the discrimination between staging groups. A larger hazard ratio indicated further distance between survival curves, and its *p*-value reflected the statistical significance of this distance. A *p*-value of less than 0.05 was considered statistically significant. All statistical analyses were performed using SPSS Statistics for Windows, Version 24.0 (released 2016; IBM Corp., Armonk, NY, USA) except competing risk analysis for BCSS using R Version 3.3.2. 

## 5. Conclusions

In conclusion, the AJCC’s 8th edition provided more accurate prognostication for the clinical prognostic stage than the anatomic stage in two independent databases of IBC patients. This suggests that incorporating biological factors into the traditional staging system provided accurate prognostic information. The clinical prognostic stage could be used in daily clinical practice for patients with IBC and to design clinical trials categorizing patients according to risk. Because there is still room for improvement in the IBC staging system, we plan to create a new IBC-specific model based on these study results.

## Figures and Tables

**Figure 1 cancers-12-03105-f001:**
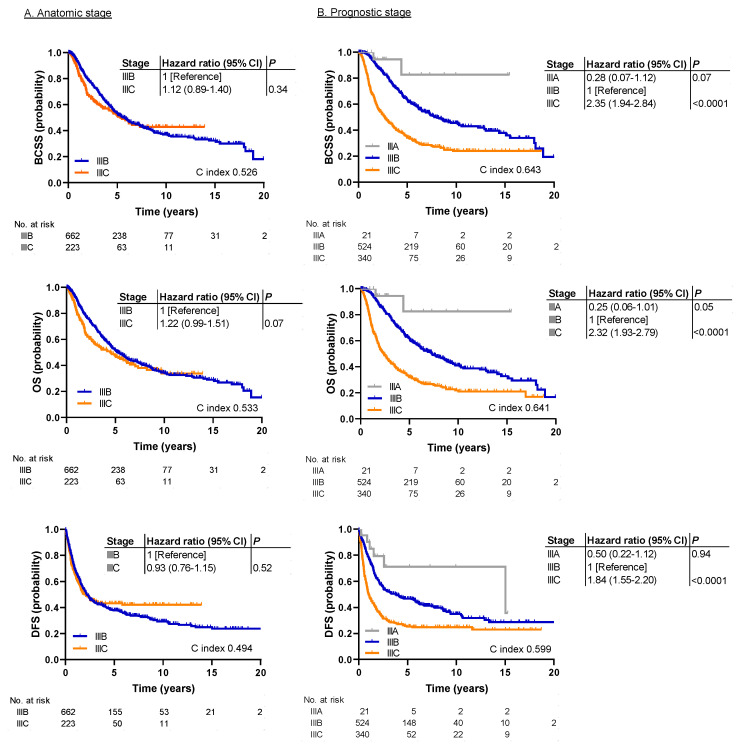
Kaplan-Meier survival plots demonstrating the association between the anatomic/prognostic staging systems and survival outcomes in the patients treated at The University of Texas MD Anderson Cancer Center (MDA cohort). (**A**) Survival analysis by anatomic stage for BCSS, OS, and DFS. No statistically significant differences were observed. (**B**) Survival analysis by prognostic stage for BCSS, OS, and DFS. There were significant differences for all endpoints. BCSS, breast cancer-specific survival; DFS, disease free survival; HR, hazard ratio; OS, overall survival.

**Figure 2 cancers-12-03105-f002:**
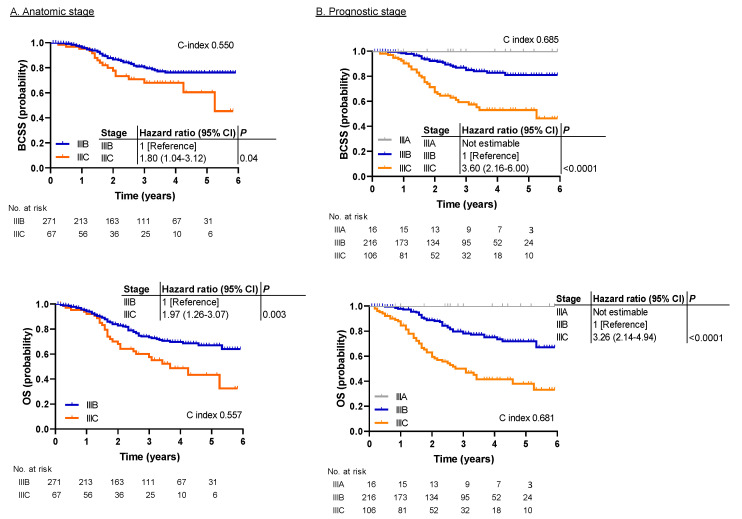
Kaplan–Meier survival plots demonstrating the association between the anatomic/prognostic staging systems and survival outcomes in patients registered in the national Surveillance, Epidemiology, and End Results (SEER) database (SEER cohort). (**A**) Survival analysis by anatomic stage for BCSS and OS. Statistically significant differences were observed in BCSS and OS. (**B**) Survival analysis by prognostic stage for BCSS and OS. Significant differences between IIIB and IIIC were more prominent with prognostic stage than with anatomic stage. BCSS, breast cancer-specific survival; HR, hazard ratio; OS, overall survival.

**Figure 3 cancers-12-03105-f003:**
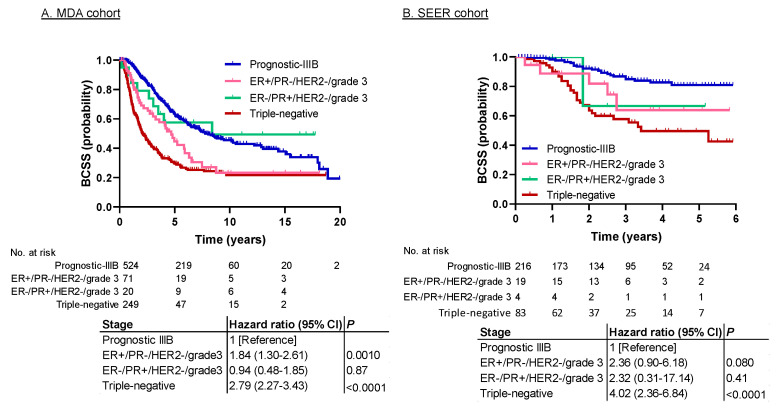
Survival analysis of prognostic stage IIIB and subgroups of prognostic stage IIIC for BCSS in the (**A**) MDA cohort and the (**B**) SEER cohort. The BCSS for ER+/PR−/HER2−/grade 3 and ER−/PR+/HER2−/grade 3 was better than that for triple-negative IBC and worse than that for prognostic stage IIIB in both cohorts. Abbreviations: BCSS, breast cancer-specific survival; ER, estrogen receptor; HER2, human epidermal growth factor receptor 2; PR, progesterone receptor.

**Figure 4 cancers-12-03105-f004:**
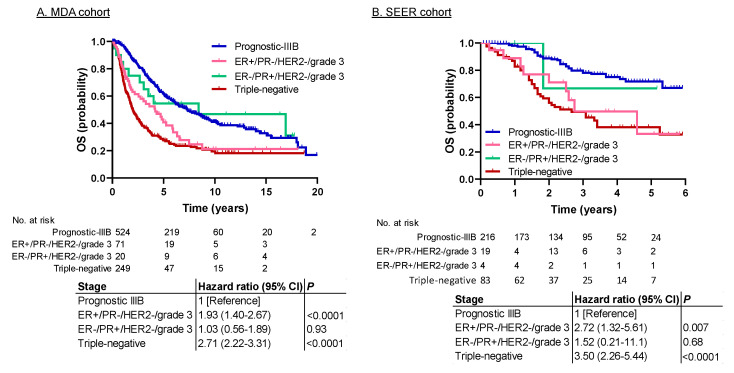
Survival analysis of prognostic stage IIIB and subgroups of prognostic stage IIIC for OS in the (**A**) MDA cohort and the (**B**) SEER cohort. The OS for ER+/PR−/HER2−/grade 3 and ER−/PR+/HER2−/grade 3 was better than that for triple-negative IBC and worse than that for prognostic stage IIIB in both cohorts. Abbreviations: ER, estrogen receptor; HER2, human epidermal growth factor receptor 2; OS, overall survival; PR, progesterone receptor.

**Table 1 cancers-12-03105-t001:** Patient characteristics.

Characteristics	MDA Cohort (*n* = 885)		SEER Cohort (*n* = 338)	
	No.	%	No.	%
Age at diagnosis in years				
Median (range)	50 (19–96)		57 (25–93)	
<40	161	18.2	34	10.1
40–49	255	28.8	60	17.8
50–59	299	33.8	104	30.8
60–69	132	14.9	78	23.1
≥70	38	4.3	62	18.3
Race				
White	679	76.7	279	82.5
Black	78	8.8	33	9.8
Other	128	14.5	26	7.7
Tumor grade				
1	8	0.9	9	2.7
2	183	20.7	105	31.1
3	694	78.4	224	66.3
ER status				
Positive	418	47.2	185	54.7
Negative	467	52.8	153	45.3
PR status				
Positive	306	34.6	145	42.9
Negative	579	65.4	193	57.1
HER2 status				
Positive	325	36.7	128	37.9
Negative	560	63.3	210	62.1
Subtype				
HR+/HER2−	311	35.1	127	37.6
HR+/HER2+	137	15.5	70	20.7
HR−/HER2+	188	21.2	58	17.2
HR−/HER2−	249	28.1	83	24.6
Lymph node stage				
N0	136	15.4	58	17.2
N1	436	49.3	140	41.4
N2	86	9.7	73	21.6
N3	227	25.6	67	19.8
AJCC anatomic stage *				
IIIB	662	74.8	271	80.2
IIIC	223	25.2	67	19.8
AJCC prognostic stage *				
IIIA	21	2.4	16	4.7
IIIB	524	59.2	216	63.9
IIIC	340	38.4	106	31.4
Chemotherapy				
Yes	863	97.5	Not known	
Neoadjuvant	834	94.3	Not known	
Adjuvant	29	3.3	Not known	
No	22	2.5	Not known	
Radiation therapy				
Yes	736	83.2	Not known	
No	149	16.8	Not known	
Treatment				
Chemotherapy + surgery + radiation	724	81.8	Not known	
Chemotherapy + surgery	139	15.7	Not known	
Surgery + radiation	12	1.4	Not known	
Surgery alone	10	1.1	Not known	

* Stages were determined on the basis of the AJCC Cancer Staging Manual, 8th edition. Abbreviations: AJCC, American Joint Committee on Cancer; ER, estrogen receptor; HER2, human epidermal growth factor receptor-2; N, node; PR, progesterone receptor.

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
