# Peer review of "Validation of Prognostic Stage and Anatomic Stage in the American Joint Committee on Cancer 8th Edition for Inflammatory Breast Cancer"

_cancers, 2020, doi:10.3390/cancers12113105_

Round 1
Reviewer 1 Report
Very nice study with a clear question, well conducted and described.
Few minor comments:
Due to the clinical confusion on the definition of inflammatory breast cancer, authors are encouraged to include a clear statement on MDA Anderson definition of IBC at diagnosis.
Did the surgery always include mastectomy?
C index should briefly be introduced and defined in the introduction, in order to facilitate reading of the paper. In the statistics section the meaning of the Harrel concordance index should be defined, and stated what exactly does the test demonstrate.
Reviewer 2 Report
This is an important article with a focus on the validation of AJCC 8th prognostic staging. The authors used two solid datasets.
However there is currently AJCC 9th staging system, hence this manuscript is a little bit lack of novelty. Also, the authors didn't provide details of patient selection, this is usually presented as a flowchart.
If the authors could address these two questions, this manuscript could be accepted.
Reviewer 3 Report
The manuscript is of interest since it is true that there is a lack of prognostic factors in inflammatory breast cancer.
The major problem is the way IBC tumors have been classified as stage IIIA, IIIB and IIIC. For example ER+/PR+/HER2+ grade 1-2 were staged IIIA while according to the AJCC 8th, grade 2 T4 tumors are staged IIIB. This is an important point because there are already only 21 tumors if we take into account grade 2…As well, I think that triple-negative grade 2 T4 tumors should be classified as IIIC and not IIIB as in this report.
Perhaps the histological grade used here as prognostic factor is in fact the nuclear grade? If so, this should be mentioned. Or some histological grades are evaluated on surgical samples post-neoadjuvant chemotherapy? I don’t think it should be done.
It is unclear whether any of the IBC tumors in the MD Anderson cohort were not themselves used to establish the revised AJCC 8th classification in 2017? Indeed, this prognostic AJCC was established on the basis of a MD Anderson series of patients treated between January 1997 and December 2006 and then extended for HER2+ to patients treated between January 2006 and December 2013. It seems difficult to validate the same classification on the ‘same’ population…
Not having the treatments received in the SEER series is a major limitation. Moreover, T4d breast cancers in the SEER cohort may have been misdiagnosed, as the diagnostic criteria for IBC have evolved. Only tumors that are clinically T4d should be considered as IBCs.
Round 2
Reviewer 3 Report
Thanks to the authors for their responses.
I really think this manuscript should be published in the journal. As I said in my first review, we need prognostic factors in IBC, and the classification proposed here works very well.
However, I still maintain that the table proposed in the response to the first comment is not in the AJCC staging Manual 8th edn. Unless I am mistaken, this table has not been published as such. The authors mention that it is derived from the manual. This can only be true since IBC were excluded from the AJCC 8th classification (as confirmed by the authors themselves). Should this table be considered as an addendum to the manual, specific to IBC? If so, the authors must clearly mention it and 'publish' this table along with the manuscript.
Author Response
Response to Reviewer 3’s Comments
Thank you so much for your comments. Please see our responses as follows.
Point 1:
“I still maintain that the table proposed in the response to the first comment is not in the AJCC staging Manual 8th edition. Unless I am mistaken, this table has not been published as such. The authors mention that it is derived from the manual. This can only be true since IBC were excluded from the AJCC 8th classification (as confirmed by the authors themselves). Should this table be considered as an addendum to the manual, specific to IBC? If so, the authors must clearly mention it and 'publish' this table along with the manuscript”
Answer:
The following table for the clinical prognostic staging system has been published in the AJCC staging manual 8th edition. The attachment is the latest version of the manual. The following staging table is on page 79 in the attached manual.
This staging table is for T4 N0-3 cases, including IBC, but not specific to IBC.
This prognostic staging table was initially created based on the non-IBC prognostic data, while this staging table is to be used for all cases, including IBC.
Therefore, it is clinically crucial to validate this staging table's prognostic power for IBC because it has not been validated yet.
